# Assessment of drug-related problems among breast cancer patients in a cancer specialty center in Nepal

Aman Kumar Sah[1,2]*, Roshan Prajapati[3], Nabin Pathak[4,5]*, Sushil Panta[2], Shila Gurung[2]*

1 Department of Hospital Pharmacy, Madhesh Institute of Health Sciences, Provincial Hospital, Janakpurdham, Madhesh Province, Nepal, 2 Pharmaceutical Sciences Program, Faculty of Health Sciences, School of Health and Allied Sciences, Pokhara University, Pokhara, Gandaki Province, Nepal, 3 Department of Medical Oncology, Bhaktapur Cancer Hospital, Bhaktapur, Bagmati Province, Nepal, 4 Drug Information Unit and Pharmacovigilance Cell, Hospital Pharmacy Department, Hetauda Hospital, Madan Bhandari Academy of Health Sciences, Hetauda, Bagmati Province, Nepal, 5 Department of Pharmacy, Madan Bhandari Academy of Health Sciences, Hetauda, Bagmati Province, Nepal

* aman.bch2021@gmail.com (AKS); pathaknabin89@gmail.com (NP); gshila@gmail.com (SG)

## Abstract

### Introduction

Breast cancer is the leading cancer among women globally, and its management typically involves surgery, radiotherapy, and chemotherapy. This combination of complex treatment increases the risk of drug-related problems (DRPs), which can negatively impact the quality of life as it prolongs the hospital stays, increases healthcare costs, and leads to morbidity and mortality. Research on DRPs in the context of breast cancer patients in Nepal is limited; therefore, this study aimed to identify DRPs, assess their prevalence, and examine their associated factors, while emphasizing the importance of pharmaceutical care.

### Methods

A cross-sectional study was conducted at Bhaktapur Cancer Hospital, Nepal, from February to May 2024. The medical records of 92 patients were reviewed to identify DRPs using the Pharmaceutical Care Network Europe V9.1 tool, along with related scientific literatures and evidence-based guidelines, and verified by two independent medical oncologists. Descriptive statistics were used to summarize the patient characteristics, while the chi-square test was used to evaluate the associations between socio-demographic and clinical characteristics with the presence of DRPs, with a p-value <0.05 considered statistically significant.

**Data availability statement:** All relevant data are attached to Supporting information files.

**Funding:** The author(s) received no specific funding for this work.

**Competing interests:** The authors have declared that no competing interests exist.

## Results

DRPs were identified in 91 of 92 patients (98.9%), totaling 104 events. Most DRPs (87.5%) concerned treatment safety, while 12.5% involved treatment effectiveness. Patient-related factors (43.4%) and drug selection (33.3%) were primary causes. Common adverse effects included alopecia (88.0%) and anorexia (84.8%). A significant association was observed between adverse drug reactions (ADRs) and DRPs (*Fisher's exact p-value = 0.011*).

## Conclusion

DRPs were highly prevalent among breast cancer patients, mainly due to safety issues, with ADRs significantly contributing to it. Findings highlight the need for larger multicentric and cohort studies, and integration of oncology pharmacy services to optimize pharmaceutical care and minimize DRPs.

## Introduction

Globally, breast cancer is the most prevalent form of cancer and is a significant cause of death in low-and middle-income countries (LMICs) [1]. The Global Cancer Observatory (GLOBOCAN) 2022 study reported 20 million new cancer cases, with 2.3 million cases of breast cancer, making it the commonly diagnosed cancer among females. Within Southeast Asia, the incidence and mortality rates are 41.7% and 14.8%, respectively [2]. In Nepal, 22,008 new cases of cancer and 14,708 cancer deaths from breast cancer (10.2% of new cases) were projected among Nepali women [3]. According to Nepal's 2018 population-based cancer registry, breast cancer accounted for 22.9% of all cases, with an age-adjusted incidence rate of 21.5 per 100,000 people [4].

Breast cancer arises from cancerous cells in the breast tissue, typically originating from the inner lining of milk ducts or milk-producing lobules [5]. Its management involves a multimodal strategy including surgery, radiotherapy, and chemotherapy, aiming for maximum therapeutic efficacy while minimizing the toxic effects. However, chemotherapy is often associated with drug-related problems (DRPs) [6,7]. Cancer patients often have coexisting diseases, and the complexity of chemotherapy, with its narrow therapeutic index and highly toxic drugs, further increases the potential for the acquisition of DRPs [8]. DRPs are linked to reduced quality of life, longer hospital stays, higher healthcare costs, and increased risk of morbidity and mortality [9,10].

The Pharmaceutical Care Network Europe (PCNE) defines DRP as an "*event or circumstance involving drug therapy that actually or potentially interferes with desired health outcomes*" [11]. In LMICs such as Nepal, many breast cancer patients seek treatment in later stages, thus leading to complex interventions and increased mortality [12]. Literature from Ethiopia shows a considerable prevalence of DRPs among breast cancer patients with adverse drug reactions (ADRs) as the highest DRPs with predictors such as neoadjuvant chemotherapy and presence of comorbidities

[13]. The medication use process in the context of cancer patients becomes complex, thereby increasing the risk among patients and significantly increasing medication errors in both pediatric and adult settings, too [14,15]. These medication errors are further responsible for DRPs [14]. This ultimately underscores the need for identification and characterization of DRPs such that, it will help in designing effective interventions to mitigate problems related to drug therapy, thereby maximizing patient care and promotion of pharmaceutical care [10].

Within the Nepalese context, literature on the identification and characterization of DRPs in breast cancer patients, along with their associated factors, remains limited due to a lack of documentation. Therefore, to advance the concept of clinical pharmacy practice, this study sought to identify and characterize DRPs, assess their prevalence, and examine the factors associated among breast cancer patients at a cancer specialty hospital in Nepal. Understanding these DRPs and their contributing factors can enhance patient care and support the continuous improvement of healthcare practices. Moreover, the findings of this study can assist the hospital in developing its own protocol and guidelines related to drug safety and pharmacy practice, ultimately contributing to the overall advancement of cancer care.

## Methods

### Study design and study site

A cross-sectional study was conducted from February to May 2024 in the Department of Medical Oncology at Bhaktapur Cancer Hospital (BCH), Dudhpati, Bhaktapur, Nepal. The design was selected to assess study variables at a single time point, without the need for longitudinal follow-up. Given the six-month duration of the M. Pharm (Clinical Pharmacy) thesis, the cross-sectional approach was appropriate for determining prevalence and exploring associations between variables. BCH is a 125-bed government-funded cancer specialty hospital that provides chemotherapy, radiotherapy, surgery, and palliative care services to approximately 20,000 cancer patients every year [16]. It serves a diverse patient population from across Nepal, encompassing varied demographic and socio-economic groups. BCH was selected as the study site because it is one of the country's leading comprehensive oncology centres, receiving referrals from multiple provinces and thereby offering access to patients with a wide spectrum of malignancies. The high case load and broad coverage make it an ideal setting to capture representative data on cancer patients in Nepal. In addition, as a government-funded institution within the capital's periphery, BCH ensures equitable access to essential medicines and services, minimizing financial barriers to care. Its established infrastructure for clinical training and research for medical students further supported the feasibility and reliability of conducting this study within the available time-frame. This study was reported in accordance with the Strengthening of the Reporting of Observational Studies in Epidemiology (STROBE) guidelines [17] (S1 File. STROBE guidelines).

### Inclusion and exclusion criteria

Female patients aged 18 years and older with a confirmed diagnosis of breast cancer who underwent at least one cycle of chemotherapy, regardless of their staging and whether the chemotherapy course was completed within the BCH medical oncology department, were included in the study. Patient refusing to participate, pregnant women, patients with psychiatric disorders, and patients who revisited the hospital for the next cycle of chemotherapy procedure were excluded. Patients were selected based on the criteria at their first meeting with the researcher, regardless of whether they were new or returning, to avoid data duplication, as the study aimed to assess DRPs per patient rather than per treatment cycle.

### Sampling size and sampling technique

Sample size of the study population was calculated considering a 50% prevalence and 5% margin of error at a 95% confidence interval (CI) using a single population formula:

$$\text{Sample size for an infinite population (n)} = z^2\frac{p\,(1-p)}{d^2} = 1.96^2\frac{0.5\,(1-0.5)}{0.05^2} = 384$$

Here, $Z$ is the critical value for a 95% CI (1.96 from the Z-table). $p$ is the proportion of DRPs in breast cancer patients and $d$ is the margin of error (5%). Since the prevalence of DRPs in breast cancer patients in Nepal is unknown, $p$ is assumed to be 50%.

The source population (N) was calculated by reviewing the medical records of breast cancer patients visiting both the inpatient and outpatient oncology units of BCH for the last three months (17th July to 16th October 2023), which was found to be 121. However, with the study population being less than 10,000, the sample size was further estimated via the following reduction formula [8]:

Corrected sample size $= \frac{n \times N}{n+N} = \frac{384 \times 121}{384+121} = 92$. Where N = source population and n = estimated sample size for N ≥ 10000.

Therefore, 92 breast cancer patients were selected for the study via a purposive sampling technique. Since, the sample size calculated was very less and the prevalence of patients coming to the hospital for breast carcinoma care is quite less, it limited our study in using the random sampling technique. As a consequence, we chose purposive sampling as it ensured proper representation of breast cancer patients with related characteristics, thus increasing the reliability of the findings. Several constraints, such as time and logistical factors, were also barriers that led us to choose purposive sampling for the study.

## Patient interview, data collection tool, and procedure

Participants were enrolled as per the criteria after receiving written informed consent. To minimize bias, strict inclusion and exclusion criteria were applied, ensuring that all selected patients had confirmed breast cancer diagnoses and complete treatment records relevant to DRPs assessment. Patient characteristics were reviewed by a clinical pharmacist (AKS) to ensure a range of ages, cancer stages, and treatment modalities, enhancing representativeness within the study population.

Required information, like socio-demographic, clinical characteristics, and medication details was recorded in the data collection sheet from the patient's record file and medication charts before they visit the doctor in the outpatient department (OPD). The participants were interviewed in a separate room after visiting the doctor in the OPD. Relevant information such as socio-demographic, clinical characteristics, medication details, including ADRs, was verified and recorded. ADRs were verified through the objective tests, physical assessment, and through the interview process with the patients; no formal causality or severity scale was applied. The DRP data were independently verified by two medical oncologists to minimize bias. The overall time duration for the data collection was about 10–15 minutes on average. In case of the missing data, the information was once again checked with the patient on the next day or when the patient visited the hospital in the day care for the next round of chemotherapy. In order to ensure data accuracy, cross-checking was conducted by the medical consultant doctors.

Following the recording of the information, DRPs were checked using the validated tool PCNE V9.1 [11]. This tool consists of domains such as problems, causes, planned intervention, intervention acceptance, and status of the DRPs, where our study was only focused on the problems and causes of DRPs [11]. The tool was piloted in 10% (10 breast cancer patients) to confirm its suitability in the Nepalese context, and no modifications were made. DRPs were identified via pertinent guidelines and literature from the National Comprehensive Cancer Network (NCCN), American Society of Clinical Oncology (ASCO), European Society for Medical Oncology (ESMO), and American Cancer Society (ACS). National-level guidelines such as Nepalese National Formulary 2018 [18] and National List of Essential Medicine 2021 [19], were also referred for the identification of DRPs. Clinical pharmacist (AKS) identified the DRPs and their associated cause after the patient received chemotherapy. If there was any confusion, then it was discussed with two independent medical oncologists from both units, i.e., I and II of the BCH Department of Medical Oncology, who verified the DRPs on the basis of practice- and evidence-based guidelines, and primary studies published in peer-reviewed journals. However, if discrepancies regarding DRPs between the oncologists were recorded, then it was further discussed in the hospital tumor board

meeting. The decision of tumor board was the final one. A total of seven disagreements between the two oncologists were resolved through discussion at a tumor board meeting, ensuring consensus on DRPs classification. The flowchart for the data collection is mentioned as Fig 1. Flowchart on data collection. Sequential steps of data collection for this study, including verification of DRPs by medical oncologists.

## Ethical consideration

Ethical approval was obtained from the Institutional Review Committee (IRC) of the Pokhara University Research Centre (PURC) [Reference No. 75/2080/81] (S2 File. Ethical approval letter). Written informed consent from the study participants was obtained before data collection to use their medical records following all the ethical rules, regulations, and standards mentioned in the Declaration of Helsinki [20]. The patient's confidentiality was ensured by following the standards as outlined in the IRC of PURC and the hospital's ethical regulations. Data were obtained and kept privately, where the patient's identifying information was made anonymous to ensure confidentiality. The study was not registered on a clinical trial registry.

## Data analysis

The data were entered into Microsoft Excel 2016 and then analyzed via the Statistical Package for Social Sciences (SPSS) Version 16. Within SPSS, the data were coded for each variable, and descriptive statistics were performed to present the data, such as socio-demographic and clinical characteristics as frequencies, percentages, and means. The data obtained were in the form of ordinal and nominal variables. Hence, to check for the association between variables,

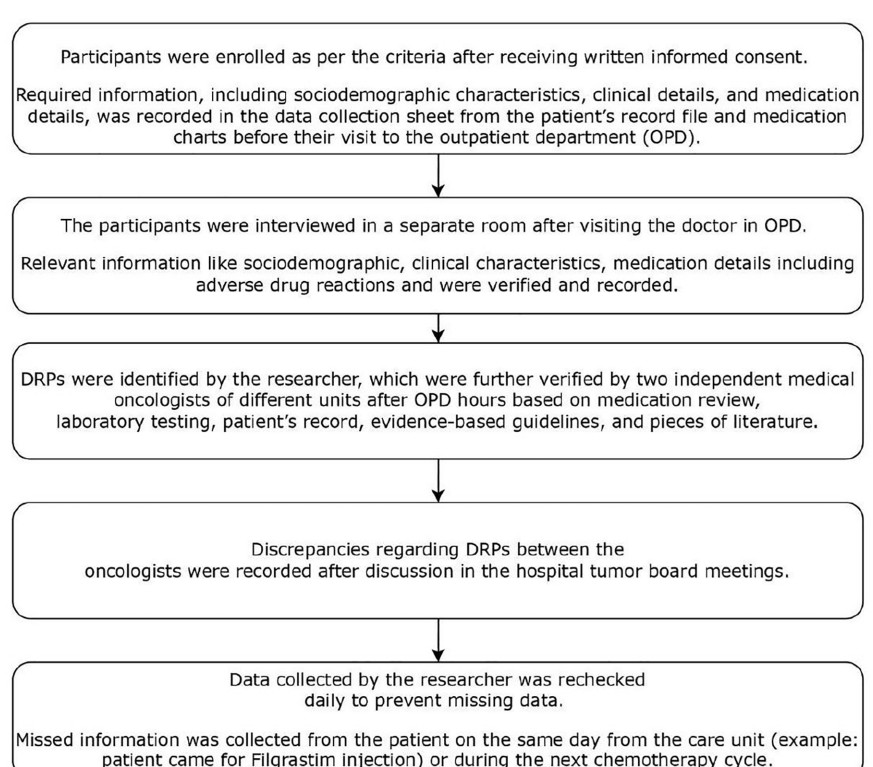

**Fig 1. Flowchart on data collection.** Describes the steps of data collection by researcher and verification of drug-related problems (DRPs) by medical oncologist.

the chi-square test was used to identify the associations between socio-demographic variables (age group, marital status, residency, education level, occupation) and clinical characteristics (stage of cancer, histological type, ER receptor status, PR receptor status, HER2 receptor status, comorbidities, family history of cancer, types of chemotherapy regimen, number of drugs prescribed, ADR, drug-drug interaction) with the presence of DRPs where a p-value of <0.05 was considered to indicate statistical significance.

## Results

### Socio-demographic and clinical characteristics of the study participants

A total of 92 female patients with breast cancer were enrolled, with a mean age of 47.25 ± 12.18 years. Among them, 55.4% were aged 36–55 years. Most patients were married (97.8%), lived in rural areas (65.2%), were illiterate (46.7%), and were housewives (60.9%).

Among the clinical characteristics, 55.4% of the patients had right breast carcinoma, and 50.0% were diagnosed with stage II breast cancer. The most common type was ductal carcinoma, which was found in 96.7% of the patients. Oestrogen receptor negativity was present in 56.5% of the patients, progesterone receptor negativity in 64.1%, and human epidermal growth factor receptor 2 (HER2) negativity in 48.9% of the patients. Furthermore, 79.3% of the patients had a Ki67 index greater than 15%.

In terms of comorbidities, 70.6% of the patients had no comorbidities, whereas hypertension was the most common comorbidity among those with comorbid conditions (14.1%). Additionally, 73.9% of the patients reported no family history of cancer. In terms of treatment, 60.9% of the patients received adjuvant chemotherapy, 86.9% were on a 3-weekly or more chemotherapy regimen, and 87.0% received five or more drugs (Table 1).

### Regimens for the management of breast cancer

The most preferred regimens included docetaxel, doxorubicin, and cyclophosphamide (TAC), which were used in 24.0% of the patients, followed by doxorubicin, cyclophosphamide, and paclitaxel (AC-T), which were used in 15.2% of the patients (Fig 2. Percentages of patients treated with different chemotherapy regimens). Each bar represents the percentage of patients treated with a specific regimen, highlighting the comparative use of different chemotherapy approaches across the patient.

### DRPs, their causes, and factors associated with DRPs

A total of 104 DRPs were identified from 91 breast cancer patients, reflecting a prevalence of 98.9% and a mean of 1.14 DRPs per patient. Among the 92 patients, 84.78% had one DRP, 14.13% had two DRPs, and 1.1% had none. Following the identification of problems as per PCNE V9.1, treatment safety (87.5%) accounted for the majority of the problems, whereas treatment effectiveness accounted for 12.5% of the total DRPs. Patient-related factors (43.4%) and drug selection (33.3%) were the major causes of DRPs in breast cancer patients (Table 2).

We have presented examples of DRPs, correlating them with the primary domains along with the comments (Table 3).

Alopecia was the most common adverse effect of chemotherapy (88.0%), followed by anorexia (84.8%) and fatigue (82.6%). More than half of the participants experienced nausea and vomiting. However, rashes were found to be the least common adverse effect among the participants enrolled in the study. Constipation, diarrhoea, headache, itching, menopause, and tastelessness were categorized as others, as they were observed in fewer cases (Table 4).

## Discussion

The present study assessed DRPs and their associated factors among breast cancer patients in a specialty cancer hospital in Nepal. Our study revealed results similar to those of studies in Ethiopia [13] and Nepal [21], where the majority

**Table 1. Socio-demographic and clinical characteristics of the study participants.**

| Socio-demographic and clinical characteristics | Category | Frequency (n=92) | Percentage (%) |
|---|---|---|---|
| Age (years) | 18-35 | 17 | 18.5 |
| | 36-55 | 51 | 55.4 |
| | 56 and older | 24 | 26.1 |
| Marital status | Single | 2 | 2.2 |
| | Married | 90 | 97.8 |
| Residency | Urban | 32 | 34.8 |
| | Rural | 60 | 65.2 |
| Education level | No formal education | 43 | 46.7 |
| | Primary | 12 | 13.1 |
| | Secondary | 30 | 32.6 |
| | Higher secondary and above | 7 | 7.6 |
| Occupation | Housewife | 56 | 60.9 |
| | Government and private employee | 7 | 7.6 |
| | Farmer | 15 | 16.3 |
| | Business | 7 | 7.6 |
| | Unemployed | 7 | 7.6 |
| Diagnosis | Left breast carcinoma | 39 | 42.4 |
| | Right breast carcinoma | 51 | 55.4 |
| | Bilateral breast carcinoma | 2 | 2.2 |
| Stage of cancer | Stage I | 3 | 3.3 |
| | Stage II | 46 | 50.0 |
| | Stage III | 25 | 27.2 |
| | Stage IV | 18 | 19.5 |
| Histological type of breast cancer | Lobular | 3 | 3.3 |
| | Ductal | 89 | 96.7 |
| Estrogen receptor status | Positive | 40 | 43.5 |
| | Negative | 52 | 56.5 |
| Progesterone receptor status | Positive | 33 | 35.9 |
| | Negative | 59 | 64.1 |
| HER2 receptor status | Positive | 41 | 44.6 |
| | Negative | 45 | 48.9 |
| | Equivocal | 6 | 6.5 |
| Ki67 index | Less than 15% | 19 | 20.7 |
| | More than 15% | 73 | 79.3 |
| Number of comorbidities present | None | 65 | 70.6 |
| | One | 23 | 25.0 |
| | Two or more | 4 | 4.4 |
| Family history of cancer | No | 68 | 73.9 |
| | Yes | 24 | 26.1 |
| Type of chemotherapy regimen | Neoadjuvant | 36 | 39.1 |
| | Adjuvant | 56 | 60.9 |
| | 2-weekly or less | 12 | 13.1 |
| | 3-weekly or more | 80 | 86.9 |
| Number of drugs prescribed | Less than 5 | 12 | 13.0 |
| | 5 or more | 80 | 87.0 |

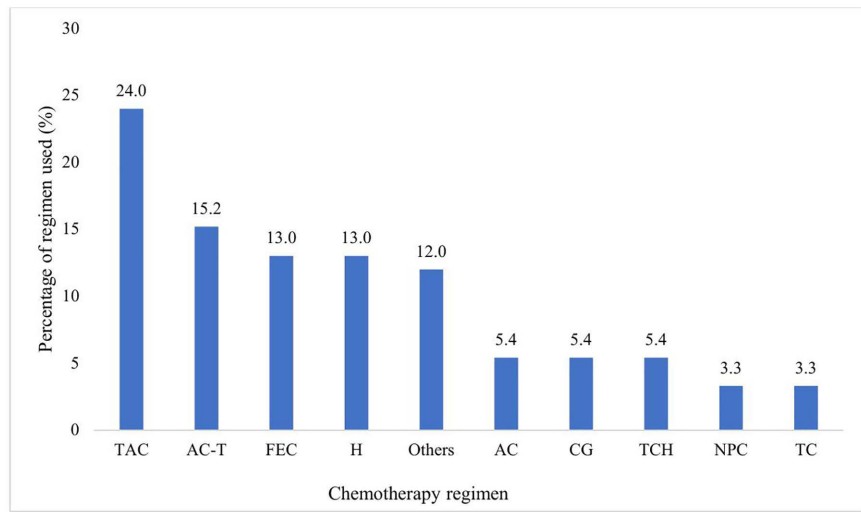

**Fig 2. Percentages of patients treated with different chemotherapy regimens.** Describes percentages of patients who received different chemotherapy regimens where, TAC: Docetaxel, doxorubicin, and cyclophosphamide; AC-T: Doxorubicin, cyclophosphamide followed by paclitaxel; FEC: Fluorouracil, epirubicin and cyclophosphamide; H, Trastzumab; others (Epirubicin & paclitaxel, paclitaxel only, Gemcitabine only, Capecitabine only, paclitaxel & Carboplatin, Trastzumab & Capecitabine, Lapatinib & Capecitabine, Doxorubicin, cyclophosphamide & paclitaxel, AC: (Doxorubicin & Cyclophosphamide), CG: (Carboplatin & Gemcitabine), TCH: (Docetaxel, Cyclophosphamide & Trastzumab), NPC: (Nanoparticle paclitaxel & Carboplatin), TC: (Docetaxel & Cyclophosphamide).

of the participants were 36–55 years old and 40–50 years old, respectively. This defines a higher probability of having breast cancer from 36–55 years of age. The majority of the participants (97.8%) were married, and more than half of the participants (65.2%) were living in rural areas. This result is similar to those studies conducted in Nepal and Kenya, where 90.0% were married out of 51 participants [22], 66.7% were married, and 66.7% lived in rural areas out of 81 participants [23]. Our study included only female breast cancer patients, as the incidence of male breast cancer is 0.44% in Nepal [24].

The majority of the participants (46.7%) were illiterate. More than half of the participants (60.9%) were housewives by occupation. This is in line with a study conducted in Ethiopia, where 65.4% and 29.9% of the participants were illiterate and housewives, respectively [13]. Most of the women included in this study were from rural areas of Nepal and were housewives by occupation.

The most common form of cancer was carcinoma of the right breast, which was present in 55.4% of the participants. Interestingly, these results are not consistent with studies conducted in Nepal and worldwide, where left breast cancer is more prevalent [21,25]. This might be because a greater number of right breast cancer patients visited the center at that time.

Histologically, ductal carcinoma (96.7%) was the predominant type of cancer. More than half of the participants (56.5%) had negative ER receptor results. Similarly, 64.1% of the participants were negative for PR receptors. The HER2 receptor was positive in 44.6% of the participants. The majority of the participants had more than 15.0% of the Ki67 index value (79.3%). This result is in line with the findings of a study conducted in Kenya, where ductal carcinoma (96.3%), ER receptor negativity (57.0%), and PR receptor negativity (57.0%) were present but inconsistent with HER2 receptor positivity (93.8%) and a Ki67 index greater than 20% [23]. This might be due to ethnic disparity in HER2-positive breast cancer, as the genetic makeup of ethnic groups differed, as mentioned by a study conducted in the USA [26]. Additionally, the Ki67 index was not recorded in 49.4% of the participants in Kenya, which might be the reason behind the variable results in their study.

**Table 2. Problem and cause domain of the PCNE classification of DRPs.**

| Domain | Code as per PCNE V9.01 | Category | Frequency (n = 104) | Percentage (%) |
|--------|------------------------|----------|---------------------|----------------|
| Problem | P1 | Treatment effectiveness | 13 | 12.5 |
| | P1.1 | No effect of drug treatment despite correct use | 1 | 1.0 |
| | P1.3 | Untreated symptoms or indication | 12 | 11.5 |
| | P2 | Treatment safety | 91 | 87.5 |
| | P2.1 | Adverse drug event (possibly) occurring | 91 | 87.5 |
| Causes | C1 | Drug selection | 43 | 33.3 |
| | C1.2 | No indication for drug | 18 | 14 |
| | C1.3 | Inappropriate combination of drugs, or drugs and herbal medications, or drugs and dietary supplements | 11 | 8.5 |
| | C1.4 | Inappropriate duplication of therapeutic group or active ingredient | 1 | 0.8 |
| | C1.5 | No or incomplete drug treatment in spite of existing indication | 11 | 8.5 |
| | C1.6 | Too many different drugs/active ingredients prescribed for indication | 2 | 1.6 |
| | C5 | Dispensing | 10 | 7.8 |
| | C5.1 | Prescribed drug not available | 4 | 3.1 |
| | C5.2 | Necessary information not provided or incorrect advice provided | 6 | 4.7 |
| | C6 | Drug use process | 1 | 0.8 |
| | C6.4 | Drug not administered at all by a health professional | 1 | 0.8 |
| | C7 | Patient-related | 56 | 43.4 |
| | C7.1 | Patient intentionally uses/takes less drug than prescribed or does not take the drug at all for whatever reason | 13 | 10.1 |
| | C7.2 | Patient uses/takes more drug than prescribed | 2 | 1.6 |
| | C7.4 | Patient decides to use unnecessary drug | 2 | 1.6 |
| | C7.7 | Inappropriate timing or dosing intervals | 38 | 29.5 |
| | C7.9 | Patient physically unable to use drug/form as directed | 1 | 0.8 |
| | C9 | Others | 19 | 14.7 |
| | C9.2 | Other cause; specify | 19 | 14.7 |

*Note: This table has been prepared in response to the PCNE V9.1. Domains/Subdomains that weren't quantified were omitted.*

The majority of the participants (70.7%) had no comorbidities, and out of the comorbidities present, hypertension was present in 14.1% of the participants, as it was the most common type. This result is slightly different from that of a study conducted in Brazil, where hypertension (61.3%) was more prevalent [27]. Very few participants (26.1%) had a family history of cancer in our study, which was similar to the findings of a study conducted in Nepal [28].

Most of the participants in our study had early-stage breast cancer (53.3%), which is different from studies conducted in Nepal, where patients were diagnosed at a late stage [29]. Screening programs for breast cancer by hospitals and government policies might have been the reason for such an early-stage diagnosis. Adjuvant chemotherapy (50.0%) was the predominant type of chemotherapy used. The duration of the chemotherapy regimen was 3-weekly or more for 86.9% of the participants. The most preferred regimen for cancer patients was TAC (23.9%). This finding is not consistent with a study conducted in Kenya, where the HER2 receptor was positive in the majority of the participants, and doxorubicin, cyclophosphamide, paclitaxel, and trastuzumab (ACTH) (80.3%) were the preferred regimens. In our study, most of the

**Table 3. Examples of DRPs.**

| Primary domain | Code as per PCNE | Cause | Examples of DRPs | Comments |
|---|---|---|---|---|
| Drug Selection | C1.2 | No indication for drug | Case 1: A 45-year-old patient diagnosed with carcinoma of the right breast post-modified radical mastectomy (MRM) received 4 cycles of AC-T. She was prescribed Lycopene, Iron, Zinc supplements, and Protein powder {containing protein in major quantity, vitamins (A, B1, B2, B3, B6, B9, B12, C, D3), Zinc, Iron, Calcium and Phosphorus in minute quantity} despite anaemia and changes in body weight at the time. | Drugs were prescribed without the need for it. |
| | | | Case 2: A 47-year-old female patient diagnosed with carcinoma of the left breast post-MRM received 5 cycles of Trastuzumab. She was prescribed Curcumin extract and apricot seed extract without the need for them. | |
| | C1.3 | Inappropriate combination of drugs, or drugs and herbal medications, or drugs and dietary supplements | Case 1: A 58-year-old female patient diagnosed with carcinoma of the right breast received 2 cycles of Fluorouracil, Epirubicin and Cyclophosphamide (FEC). While undergoing chemotherapy, she visited an ayurvedic doctor and was prescribed ayurvedic medicines, including *Triphala churna (Haritaki, Bibhitaki, and Amalaki)*, Ashwagandha (Withania) powder, and Curcumin powder. | The combination of the drugs was inappropriate, as the patient complained of insomnia and severe fatigue. |
| | C1.4 | Inappropriate duplication of therapeutic group or active ingredient | Case 1: A 72-year-old female with carcinoma of the right breast post-MRM received 1 cycle of FEC. She was prescribed Pantoprazole and Rabeprazole simultaneously. | Prescribing the drug from the same therapeutic class resulted in inappropriate duplication. |
| | C1.5 | No or incomplete drug treatment in spite of existing indication | Case 1: A 47-year-old female diagnosed with carcinoma of the right breast post-MRM received 2 cycles of FEC. She suffered from pain near the right breast area and upper right limb, which remained untreated. | No medications were provided to the patient, despite the existing symptoms. |
| | | | | Case 2: A 46-year-old patient diagnosed with carcinoma of the left breast received 3 cycles of Docetaxel, Cyclophosphamide, and Trastuzumab (TCH). She suffered from itching of the upper limb, which remained unaddressed. |
| | C1.6 | Too many different drugs/active ingredients prescribed for indication | Case 1: A 26-year-old female diagnosed with carcinoma of the right breast post-MRM received 2 cycles of Paclitaxel, Cyclophosphamide, and Doxorubicin. She was prescribed Apricot seed extract, grape seed, and green tea extract, Protein powder {containing protein in major quantity, vitamins (A, B1, B2, B3, B6, B9, B12, C, D3), Zinc, Iron, Calcium and Phosphorus in minute quantity}, oral Chlorhexidine gargle, with a Metoclopramide tablet. | Too many drugs/supplements were prescribed for the indications. |
| Dispensing | C5.1 | Prescribed drug not available | Case 1: A 35-year-old female diagnosed with carcinoma of the left breast received 2 cycles of TCH. She was prescribed Trastuzumab, which was not available in the market. As a result, the patient missed the drug during the first chemotherapy, although other chemotherapies were administered from the beginning. | The patient couldn't take the chemotherapy because of its unavailability. |
| | | | Case 2: A 76-year-old female diagnosed with carcinoma of the left breast post-MRM received 7 cycles of Paclitaxel. She did not receive the regular supply of chemotherapy from the hospital pharmacy, even though it was included in the national health insurance plan. | |
| | C5.2 | Necessary information not provided or incorrect advice provided | Case 1: A 40-year-old female diagnosed with carcinoma of the right breast received 1 cycle of TAC. She drank 10 ml of Chlorhexidine gargle because the necessary information about its use was not provided by the pharmacy. | Necessary information about the administration of drugs was not provided from the pharmacy. |

*(Continued)*

| Primary domain | Code as per PCNE | Cause | Examples of DRPs | Comments |
|---|---|---|---|---|
| Drug use process | C6.4 | Drug not administered at all by a health professional | Case 1: A 66-year-old female diagnosed with carcinoma of the left breast with lung metastasis received 1 cycle of TAC. She was prescribed Pegylated filgrastim for neutropenia, but the nurse did not administer the drug due to the overcrowding of patients in the day-care unit. The patient received the drug the next day. | Drugs were not administered by nursing staff. |
| Patient related | C7.1 | Patient intentionally uses/takes less drug than prescribed or does not take the drug at all for whatever reason | Case 1: A 31-year-old female diagnosed with carcinoma of the left breast post-MRM with lymph node metastasis received 3 cycles of Carboplatin and Gemcitabine (CG). She returned to the hospital for the 4th cycle of chemotherapy only after 2 months because of some personal issues. The chemotherapy regimen was then changed to TAC as the disease had progressed and spread to the lymph nodes. | The patient intentionally takes less drug or does not take drugs for personal reasons. |
| | | | Case 2: A 39-year-old female diagnosed with carcinoma of the left breast post-MRM received 17 cycles of Trastuzumab. She was prescribed Rabeprazole, Protein powder {containing protein in major quantity, vitamins (A, B1, B2, B3, B6, B9, B12, C, D3), Zinc, Iron, Calcium and Phosphorus in minute quantity}, Iron and Lycopene supplements, but she decided not to take Rabeprazole, which was prescribed for her. | |
| | C7.2 | Patient uses/takes more drug than prescribed | Case 1: A 39-year-old female diagnosed with recurrent carcinoma of the right breast received 3 cycles of Nanoparticle Paclitaxel and Carboplatin (NPC). She took a Ranitidine tablet (self) along with Rabeprazole (prescribed by her doctor) when she felt abdominal discomfort. | The patient takes more medicine than prescribed. |
| | C7.4 | Patient decides to use unnecessary drug | Case 1: A 40-year-old female patient diagnosed with carcinoma of the right breast with lung and bone metastasis received 2 cycles of NPC. She took immunity booster (Ganoderma lucidum) supplements from Vestige. | The patient decides to take unnecessary drugs/supplements. |
| | C7.7 | Inappropriate timing or dosing intervals | Case 1: A 52-year-old female diagnosed with metastatic breast carcinoma has received 12 cycles of Docetaxel and Cyclophosphamide (TC) regimen. She was prescribed a Rabeprazole tablet and a Domperidone tablet, which should be taken in an empty stomach 30–60 minutes before food. Occasionally, she took the medication 5–10 minutes before eating. | The timing of drug intake was inappropriate. |
| | | | | Case 2: A 43-year-old female diagnosed with carcinoma of the right breast with bone metastasis was prescribed a Letrozole tablet. She was unable to take the medication at the same time each day. |
| | C7.9 | Patient physically unable to use drug/form as directed | Case 1: A 65-year-old female diagnosed with carcinoma of the right breast received 3 cycles of Epirubicin and Docetaxel. She was prescribed Iron and Lycopene supplements which were larger capsules. | The patient found it difficult to swallow the capsules due to their larger size. |
| Others | C9.2 | Other cause; specify | Case 1: A 48-year-old female patient diagnosed with carcinoma of the right breast post-MRM received 10 cycles of Trastuzumab. Due to the high cost of *Vivitra*, the treatment was switched to the cheaper brand, *Eleftha*. The patient subsequently suffered from rashes and a fever. | The cost of chemotherapy also contributed to drug-related problems, categorized under others. |
| | | | Case 2: A 27-year-old female patient diagnosed with carcinoma of the right breast received one cycle of TAC. She was HER2 positive, and Trastuzumab was the recommended drug. However, due to the high cost of immunotherapy, the patient refused it and instead underwent TAC regimen, which resulted in adverse effects such as neutropenia, nausea, vomiting, anorexia, and fatigue. | |

**Table 4. Adverse effects of chemotherapy.**

| Adverse effects | Frequency (n = 92) | Percentage (%) |
|---|---|---|
| Alopecia | 81 | 88.0 |
| Anaemia | 41 | 44.6 |
| Nausea | 70 | 76.1 |
| Vomiting | 57 | 62.0 |
| Anorexia | 78 | 84.8 |
| Fever | 32 | 34.8 |
| Pain | 36 | 39.1 |
| Neutropenia | 15 | 16.3 |
| Neuropathy | 9 | 9.8 |
| Fatigue | 76 | 82.6 |
| Rashes | 7 | 7.6 |
| Others | 35 | 38.0 |

A statistically significant association between ADRs and the presence of DRPs was observed (Table 5). No significant associations were found between other variables and DRPs.

participants had stage II cancer, the HER2 receptor was negative, and the TAC regimen was preferred because of its better efficacy than other regimens in patients with adjuvant, triple-negative, and/or HER2-negative breast cancer [30–33].

Although guidelines recommend a defined set of drugs for breast cancer management, our study observed a higher number of prescriptions, including multivitamins, suggesting potential overprescribing. The majority of the participants (86.9%) received five or more drugs, consistent with findings from Ethiopia, where over 91.0% of patients were treated with similar polypharmacy [13]. This pattern is likely explained by the complexity of the chemotherapy regimens and the use of adjuvant drugs to manage chemotherapy-induced ADRs [34].

Our study revealed a 98.9% prevalence of DRPs, which is alarmingly high at first glance. Multiple studies revealed an average of 2–3 DRPs per patient [13,27], whereas our study revealed an average of 1.14 DRPs per patient among 104 DRPs, which were identified in 91 patients. However, efforts are still necessary to reduce the number of DRPs per patient to ensure patient safety and enhance clinical outcomes. Several reports from low-resource or high-risk clinical environments, particularly those involving polypharmacy, limited clinical pharmacy services, or high patient acuity, have documented DRP prevalence rates exceeding 90% [7]. This suggests that our findings, while high, are not unprecedented. DRPs were assessed with the validated PCNE V9.1 tool by trained professionals using predefined criteria to minimize subjectivity. As classification systems differ in sensitivity, prevalence estimates are influenced by both their granularity and the assessor's training. Standardized tools and structured training are therefore essential to ensure reliable and comparable findings.

Following the identification of problems as per the PCNE V9.1 tool, the majority of the issues were associated with treatment safety (87.5%), which contains the subdomain of adverse drug events (possibly) occurring, followed by treatment effectiveness, where the untreated symptoms or indicated symptoms (11.5%) dominated over all the other subdomains. ADRs also affect the health-related quality of life and negatively impact the patient-related outcomes, thus underscoring the need for identifying the patient's underlying health conditions, along with supportive treatment such as social and emotional support [35,36]. Huge prevalence of such DRPs also requires prompt response from the pharmacovigilance section within the hospital to prevent and help report any case of unprecedented events in the future [37]. Although the Department of Drug Administration (DDA) of Nepal supports hospital-based pharmacovigilance (PV) units, practical implementation is limited. BCH, which currently lacks an ADR reporting system, faces challenges in establishing a PV centre. With strong institutional commitment and adequate resources, BCH could develop a regional PV centre to enhance the monitoring and management of DRPs.

**Table 5. Factors responsible for DRPs.**

| Factors | Category | DRP | | p-value |
|---|---|---|---|---|
| | | No | Yes | |
| Age groups | 18-35 years | 0 | 17 | 0.257 |
| | 36-55 years | 0 | 51 | |
| | 56 years & older | 1 | 23 | |
| Marital status | Single | 0 | 2 | 1 |
| | Married | 1 | 89 | |
| Residency | Urban | 0 | 32 | 1 |
| | Rural | 1 | 59 | |
| Education level | Illiterate | 1 | 42 | 0.467 |
| | Literate | 0 | 49 | |
| Occupation | Unemployed | 0 | 7 | 1 |
| | Employed | 1 | 84 | |
| Stage of cancer | Early stage | 1 | 48 | 1 |
| | Advance stage | 0 | 43 | |
| Histological types | Lobular | 0 | 3 | 1 |
| | Ductal | 1 | 88 | |
| ER receptor status | Positive | 1 | 39 | 0.435 |
| | Negative | 0 | 52 | |
| PR receptor status | Positive | 1 | 32 | 0.359 |
| | Negative | 0 | 59 | |
| HER2 receptor status | Positive | 0 | 41 | 0.486 |
| | Negative | 1 | 44 | |
| | Equivocal | 0 | 6 | |
| Comorbidities | Absent | 0 | 65 | 0.293 |
| | Present | 1 | 26 | |
| Family history of cancer | No | 0 | 68 | 0.261 |
| | Yes | 1 | 23 | |
| Types of chemotherapy regimen | Neoadjuvant | 0 | 35 | 1 |
| | Adjuvant | 1 | 56 | |
| No. of drugs prescribed | Less than 5 | 1 | 11 | 0.13 |
| | 5 and above | 0 | 80 | |
| ADR | Absent | 1 | 0 | 0.011* |
| | Present | 0 | 91 | |
| Drug–drug interactions | Absent | 1 | 5 | 0.065 |
| | Present | 0 | 86 | |

*(Statistically significant at Fischer-Exact p-value <0.05)*

Patient-related (43.4%) domains were the primary cause of DRPs. An inappropriate timing or dosing interval (29.5%) was the major subdomain responsible for the cause of DRPs. Timing or dosing issues may stem from a combination of health literacy gaps, regimen complexity, and system failures, including resource limitations and inadequate infrastructure. This finding is not consistent with a study conducted in Brazil, where DRPs were in the category of indication (37.8%), followed by safety (23.8%). This finding indicates that the causes of DRPs vary from study to study, irrespective of the type and number of DRPs.

A significant association between ADRs and the presence of DRPs was found, whereas other socio-demographic and clinical characteristics were not significant. However, due to the cross-sectional design of the study, a causal relationship could not be determined. This finding contrasts with a study conducted in Ethiopia, where concurrent comorbidities and neoadjuvant chemotherapy were identified as factors contributing to DRPs [13]. The discrepancy may be due to the higher prevalence of ADRs in the present study population.

Alopecia (88.0%) and anorexia (84.8%) were the most common adverse effects of chemotherapy, followed by fatigue (82.6%), in the majority of the participants in our study. This finding is somewhat consistent with a study conducted among 77 cancer patients in India, where fatigue (87.0%) followed by anorexia (71.4%) were the major adverse effects of chemotherapy [38]. The fluctuation might be due to variations in the clinical characteristics of the participants as well as the treatment protocol of the study site.

Potential intervention in the coming days to resolve the DRPs could be the inclusion of health professionals, primarily clinical pharmacists, in identifying and characterizing DRPs, as several studies highlight the role of pharmacists in providing pharmaceutical care [39–45]. The hospital pharmacy service guideline 2015 of Nepal also includes a specific regulation for the inclusion of a clinical pharmacist in hospitals above 51 beds [46]. This provision could help in establishing proper clinical pharmacist placement and residency programs within the oncology settings too, such as to intervene in resolving DRPs to promote the idea of pharmaceutical care [46]. The development of a hospital's own independent guidelines, continuous medical education, continuous pharmacy development, along with the pro-activeness of the Drug and Therapeutic Committee in increasing awareness among healthcare professionals regarding the identification and characterization of DRPs is significant in mitigating DRPs and their causes. Similarly, our study shows a huge prevalence of ADRs as DRPs. This underscores the necessity for timely reporting, recording, and dissemination of ADR-related information through the establishment of a pharmacovigilance section in the hospital [37,47]. Similarly, our findings, especially with respect to prescription-related factors, could be disseminated well to the medical council to help in de-prescribing such non-evidence-based, inappropriate products.

## Strengths and limitations

To our knowledge, this was the first study from Nepal that reported on the identification and characterization of DRPs among breast cancer patients from a specialty hospital, underscoring the need to focus on pharmaceutical care and patient safety. However, this study also had some limitations. Since the study was conducted with a limited patient population, it might be difficult to generalize to a larger extent. It further limits to establish the causality between several variables. Because of the single-point measure, the nature of the study, the longitudinal effects of several independent variables were not possible to measure. No adjustment was made for potential confounders, and multivariable analysis was also not conducted. The way participants have been enrolled in this study might have also introduced sample bias, as several independent variables might have affected the sampling procedure. Similarly, the sensitivity analysis of the study was not conducted. Confidence intervals of the key finding were not included in the study. Due to the low sample size, this study further underscores the need for larger multicentric and cohort studies to confirm the findings, which are focused not only on identifying DRPs but also on applying interventions, especially in countries where the role of the clinical pharmacist has not yet been fully implemented. While the study draws major reflections from countries like Ethiopia and Kenya, limited data from other LMICs regarding DRPs in breast cancer limit the study from pointing out the cross-cultural relevance. Similarly, the study does not explain any interventional technique or tool that might have also affected DRPs prevalence. The verification by medical oncologists might have affected our prevalence of DRPs, which could have been considered.

## Conclusion

This study revealed a relatively high prevalence of DRPs in breast cancer patients visiting medical oncology outpatient departments in Nepal. Treatment safety was the major problem identified, whereas patient-related factors were the major domain responsible for the cause of DRPs. Alopecia was the most common adverse effect of chemotherapy, and a

significant association was observed between ADRs and the presence of DRPs in patients. The high prevalence of DRPs in breast cancer patients underscores the need for mitigation strategies through the inclusion of pharmacists in the clinical settings. Regular auditing of prescriptions, increasing awareness of healthcare professionals regarding the context of DRPs and their impact, is crucial in oncology settings to mitigate the future consequences of DRPs. Further, larger multi-centric and cohort studies should be conducted to identify DRPs, their predictors with interventional models.

## Supporting information

**S1 File. STROBE guidelines.**
(DOCX)

**S2 File. Ethical approval letter.**
(DOCX)

**S3 File. Data file.**
(XLSX)

## Acknowledgements

We would like to express our heartfelt gratitude to all the staff of Bhaktapur Cancer Hospital, Nepal, and the study participants for their cooperation during the study.

## Author contributions

**Conceptualization:** Aman Kumar Sah, Shila Gurung.

**Data curation:** Aman Kumar Sah, Sushil Panta.

**Formal analysis:** Aman Kumar Sah, Nabin Pathak, Shila Gurung.

**Investigation:** Roshan Prajapati.

**Methodology:** Aman Kumar Sah, Sushil Panta, Shila Gurung.

**Project administration:** Roshan Prajapati, Nabin Pathak, Shila Gurung.

**Resources:** Roshan Prajapati.

**Software:** Nabin Pathak, Sushil Panta.

**Supervision:** Roshan Prajapati, Sushil Panta, Shila Gurung.

**Validation:** Roshan Prajapati.

**Visualization:** Nabin Pathak, Sushil Panta.

**Writing – original draft:** Aman Kumar Sah, Nabin Pathak.

**Writing – review & editing:** Sushil Panta, Shila Gurung.

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
