## [Decision Letter · Decision Letter 0]

22 Apr 2025

PONE-D-24-58527Assessment of drug-related problems among breast cancer patients in a cancer specialty center in NepalPLOS ONE

Dear Dr. Sah,

Thank you for submitting your manuscript to PLOS ONE. After careful consideration, we feel that it has merit but does not fully meet PLOS ONE’s publication criteria as it currently stands. Therefore, we invite you to submit a revised version of the manuscript that addresses the points raised during the review process.

We look forward to receiving your revised manuscript.

Kind regards,

Vijayaprakash Suppiah, PhD

Academic Editor

PLOS ONE

Journal Requirements:

2. Please include captions for your Supporting Information files at the end of your manuscript, and update any in-text citations to match accordingly. Please see our Supporting Information guidelines for more information: http://journals.plos.org/plosone/s/supporting-information .

3. We note that your Data Availability Statement is currently as follows: All relevant data are within the manuscript and in Supporting Information files.

4. We are unable to open your Supporting Information file “Data.zip”. Please kindly revise as necessary and re-upload.

Reviewers' comments:

Reviewer's Responses to Questions

**Comments to the Author**

1. Is the manuscript technically sound, and do the data support the conclusions?

Reviewer #1: Partly

Reviewer #2: Yes

2. Has the statistical analysis been performed appropriately and rigorously? 

Reviewer #1: Yes

Reviewer #2: Yes

3. Have the authors made all data underlying the findings in their manuscript fully available?

Reviewer #1: Yes

Reviewer #2: Yes

4. Is the manuscript presented in an intelligible fashion and written in standard English?

Reviewer #1: No

Reviewer #2: Yes

5. Review Comments to the Author

Reviewer #1: • The title is clear and concise, accurately reflecting the study's focus on drug-related problems (DRPs) among breast cancer patients in Nepal.

• The abstract provides a good summary of the study, including the background, methods, results, and conclusion. However, it could benefit from a brief mention of the study's limitations.

• The introduction could be enhanced by providing more detailed information on previous studies related to DRPs in breast cancer patients, particularly in similar settings.

• The sample size calculation is explained, but the rationale for choosing a 50% prevalence rate could be clarified. Additionally, more information on how patient interviews were conducted and how data accuracy was ensured would be beneficial.

• The rationale for choosing a cross-sectional design could be elaborated. Additionally, more details about the hospital's patient population and its relevance to the study could provide better context.

• The manuscript could benefit from a more detailed description of how patient confidentiality was maintained and any measures taken to ensure ethical standards were upheld throughout the study.

• The exclusion criteria are not mentioned and should be included to provide a complete understanding of the study population.

• More details on how patient interviews were conducted, including the interview process, duration, and any training provided to the interviewers, would enhance the transparency of the data collection process. Additionally, the manuscript should describe how data accuracy and consistency were ensured.

• The manuscript could benefit from a more detailed explanation of the statistical analysis, including any software used and the rationale for choosing specific statistical tests. Additionally, the manuscript should discuss how missing data were handled and any sensitivity analyses performed.

• The manuscript should provide more details on the criteria used by the oncologists to verify DRPs and how discrepancies between the two oncologists were resolved.

• The Methods section should explicitly mention potential limitations related to the study design, data collection, and analysis. For example, the use of a single center and the potential for selection bias should be discussed.

• To enhance the rigor and transparency of the study, the manuscript should provide more detailed justifications for methodological choices, describe the data collection and verification processes in greater detail, and explicitly discuss potential limitations.

• The results section could be improved by providing more detailed statistical analysis, including confidence intervals for key findings. Additionally, a more in-depth discussion of the types of DRPs and their potential impact on patient outcomes would be valuable.

• The discussion could be strengthened by exploring potential interventions to reduce DRPs and improve patient outcomes. Additionally, the study's limitations should be discussed in more detail, including potential biases and the generalizability of the findings.

• The conclusion could be enhanced by providing specific recommendations for future research and clinical practice.

• Ensure all references are formatted consistently according to the journal's guidelines.

• To enhance the manuscript, consider providing more detailed statistical analysis, discussing the study's limitations more thoroughly, and offering specific recommendations for clinical practice and future research.

Reviewer #2: This is an interesting study and the authors have done a good job on its execution. Please find some of my comments for better outcome for manuscript:

Introduction section:

-The whole manuscript may be revised for clarity and flow. Redundant concepts/ideas need to be revised (Line 66-68 and 70-72: pretty much share the same idea of patients being susceptible and a lack of local research).

-Line 69-70: Consider reframing this sentence. The sentence is phrased awkwardly as of now.

-The authors may emphasize the rationale of this study more in the last section of the Introduction.

Method Section:

-Since the sample size is calculated, a clear justification for the use of the purposive sampling method (non-probabilistic sampling method) needs to be made. Was it because of a smaller population group, feasibility issues, or was it something else?

-DRP identification by two individuals was a great approach, but how was any discrepancy among them resolved? Adding some information about the management of conflicts in case of occurrence would add value.

-Explain the terms used in the formula for sample size calculations. It is not well formulated in the current version.

-If possible, information on which variables (sociodemographic or clinical variables) were tested for association with DRPs in the method section itself would add better flow to the reader.

Results and discussion:

-The results and discussion sections are well formulated. Make sure the percentages used in the brackets are aligned with the one mentioned in the result writing and tables.

-Line 135: 91 or 92 patients?

Conclusion:

-Line 245: May be rephrased by removing the word ‘significant’ since this is not regarding any statistical test, but rather just the frequency.

6. PLOS authors have the option to publish the peer review history of their article (what does this mean? ). If published, this will include your full peer review and any attached files.

**Do you want your identity to be public for this peer review?** For information about this choice, including consent withdrawal, please see our Privacy Policy .

Reviewer #1: No

Reviewer #2: No

---

## [Author Response · Author response to Decision Letter 1]

6 Jun 2025

Respected Editor in Chief,

Warm greetings,

I am writing on behalf of all the authors to submit the revised version (PONE-D-24-58527) of our manuscript, "Assessment of drug-related problems among breast cancer patients in a cancer specialty center in Nepal," for consideration for publication in the PLOS One. I appreciate the constructive feedback by the reviewers and the editorial team provided, and I believe the revisions have addressed the concerns raised.

In response to the reviewers' comments, I have carefully revised the manuscript, incorporating changes that enhance the work's clarity, rigor, and overall quality. To facilitate the review process, I have used the "Track Changes" feature in the document to highlight the modifications made.

I have attached the revised manuscript and a detailed response to each reviewer's comments. I trust that these changes align with the standards and expectations of the PLOS One and contribute to the overall improvement of the manuscript.

I am grateful for the opportunity to submit this revised version. I am confident that the modifications made have strengthened the manuscript and aligned it more closely with PLOS one aims and scope.

Thank you for considering our submission. I look forward to the opportunity to contribute to the scholarly dialogue in your esteemed journal. I am available for any further information or clarification that may be required during the review process.

Warm regards,

Aman Kumar Sah

Email ID: aman.bch2021@gmail.com

---

## [Decision Letter · Decision Letter 1]

6 Aug 2025

PONE-D-24-58527R1Assessment of drug-related problems among breast cancer patients in a cancer specialty center in NepalPLOS ONE

Dear Dr. Sah,

Thank you for submitting your manuscript to PLOS ONE. After careful consideration, we feel that it has merit but does not fully meet PLOS ONE’s publication criteria as it currently stands. Therefore, we invite you to submit a revised version of the manuscript that addresses the points raised during the review process.

Reviewer 1 has highlighted some concerns. Please try to address these concerns in the next revision. 

We look forward to receiving your revised manuscript.

Kind regards,

Vijayaprakash Suppiah, PhD

Academic Editor

PLOS ONE

Journal Requirements:

Reviewers' comments:

Reviewer's Responses to Questions

**Comments to the Author**

1. If the authors have adequately addressed your comments raised in a previous round of review and you feel that this manuscript is now acceptable for publication, you may indicate that here to bypass the “Comments to the Author” section, enter your conflict of interest statement in the “Confidential to Editor” section, and submit your "Accept" recommendation.

Reviewer #1: (No Response)

Reviewer #2: All comments have been addressed

2. Is the manuscript technically sound, and do the data support the conclusions?

Reviewer #1: Yes

Reviewer #2: Yes

3. Has the statistical analysis been performed appropriately and rigorously? 

Reviewer #1: Yes

Reviewer #2: Yes

4. Have the authors made all data underlying the findings in their manuscript fully available?

Reviewer #1: Yes

Reviewer #2: (No Response)

5. Is the manuscript presented in an intelligible fashion and written in standard English?

Reviewer #1: Yes

Reviewer #2: No

6. Review Comments to the Author

Reviewer #1: General Structure and Rationale

1. Clearly state whether the primary goal is to estimate the prevalence of DRPs or explore associated risk factors. The current phrasing oscillates.

2. The rationale is sound, but why was this particular center (BCH) chosen? More details on selection rationale beyond "governmental and accessible" would help.

3. The manuscript claims to be the first in Nepal, but doesn't reference enough regional studies to support this. Could there be unpublished or gray literature?

4. There is no reflection on how temporality or causality is limited in the discussion; should be explicitly acknowledged.

5. Repetition of phrases like “increased DRPs” and “pharmaceutical care” reduces clarity. Consider more varied academic phrasing.

Abstract and Keywords

6. The abstract includes excessive numerical detail and minor results; consider summarizing key findings more concisely.

7. Add more specific terms such as “PCNE classification,” “Nepal,” or “oncology pharmacy” for better indexing.

Methods

8. Purposive sampling is a major limitation—state how selection bias was minimized and how representativeness was ensured.

9. The initial assumption of 50% prevalence is fine for conservative estimates, but justify why 3 months of data was chosen.

10. Was the data collection tool (based on PCNE) piloted or validated in this setting?

11. Why exclude patients who returned for further chemo? This could skew the sample toward early-stage patients.

12. More information is needed on how disagreements were resolved between oncologists. What was the inter-rater reliability?

13. Were adjustments made for potential confounders (age, comorbidities, etc.)? No multivariable analysis was conducted.

14. Consider at least a basic sensitivity test for key findings due to small sample size.

15. When exactly was DRP assessed—before, during, or after chemo cycles? Clarity would improve reproducibility.

16. How were ADRs classified—was any scale (Naranjo, CTCAE) used?

17. IRB approval is cited but mention if this study was registered on any clinical trial registry.

Statistical Analysis

18. Only chi-square and Fisher’s exact tests were used. Consider logistic regression to control for multiple variables simultaneously.

19. Absence of CIs undermines the robustness of prevalence and association measures.

20. Effect sizes (odds ratios, relative risks) are not reported, which makes interpretation of significance less informative.

21. Given many comparisons, were p-values adjusted to account for Type I error?

Results

22. The 98.9% DRP prevalence is alarmingly high. Is this comparable to similar populations, or could it reflect overclassification?

23. Were any participants excluded due to incomplete data? This is not discussed.

24. At one point, 104 DRPs in 91 patients are discussed—make clearer what proportion had multiple DRPs.

25. Table 1 includes too many levels—merge low-frequency categories (e.g., occupation).

26. Reference is made to a figure not shown in the review—ensure figures are labeled and included.

Discussion

27. Discussion draws mainly on Ethiopia and Kenya—are there more LMIC examples to support cross-cultural relevance?

28. Phrases such as “DRPs were caused by…” should be tempered due to study design limits.

29. Expand on why timing/dosing issues are so prevalent—does this point to health literacy gaps, regimen complexity, or system failures?

30. High number of drugs prescribed is noted—discuss whether this is guideline-recommended or potentially excessive.

31. Many intervention ideas are proposed, but no feasibility or pilot data are cited—frame as hypotheses or recommendations.

32. Good point raised—but explain if Nepal’s national PV system supports hospital-based units practically.

33. Does BCH have a formal reporting mechanism for ADRs? Mention if findings have been/will be reported there.

Language and Style

34. Past and present tenses are used inconsistently throughout the methods and results.

35. Phrases like “it might be difficult to generalize the findings” occur multiple times. Use once and move on.

36. Numerous minor grammatical errors (e.g., "do not explain" should be "does not explain")—recommend copyediting pass.

37. "Protein powder," "immunity booster"—provide generic names or ingredient lists to ensure scientific precision.

38. Terms like “TAC” and “FEC” appear before being defined—define all chemotherapy abbreviations upon first use.

Conclusion

39. Statements about fulfilling National Cancer Control Strategy goals are too broad. Keep the focus on DRP assessment and interventions.

40. Mention specific designs (e.g., prospective cohort, pharmacist-led intervention RCTs) instead of vague calls for “more studies.”

Reviewer #2: So far my comments has been addressed but I highly recommend the authors to use some sort of tools to improve the understanding of the sentences in manuscript. I still stick with the comment for revising the writing/structure of the sentences throughout the manuscript to make it unambiguous with appropriate flow.

7. PLOS authors have the option to publish the peer review history of their article (what does this mean? ). If published, this will include your full peer review and any attached files.

**Do you want your identity to be public for this peer review?** For information about this choice, including consent withdrawal, please see our Privacy Policy .

Reviewer #1: **Yes: ** Sunil Shrestha

Reviewer #2: No

---

## [Author Response · Author response to Decision Letter 2]

1 Sep 2025

Date: 01/09/2025

Editor in Chief,

PLOS One,

Subject: Response to Reviewers to our Manuscript [PONE-D-24-58527R1]

Respected Editor in Chief,

Warm greetings,

I am writing to submit the second revised version (PONE-D-24-58527R1) of our manuscript, " Assessment of drug-related problems among breast cancer patients in a cancer specialty center in Nepal," for consideration for publication in PLOS One. I appreciate the constructive feedback from the reviewers and the editorial team provided, and I believe the revisions have addressed the concerns raised.

In response to the reviewers' comments, I have carefully revised the manuscript, incorporating changes that enhance the work's clarity, rigor, and overall quality. To facilitate the review process, I have used the "Track Changes" feature in the document to highlight the modifications made.

I have attached the revised manuscript and a detailed response to each reviewer's comments. I trust that these changes align with the standards and expectations of PLOS One and contribute to the overall improvement of the manuscript.

I am grateful for the opportunity to submit this revised version. I am confident that the modifications made have strengthened the manuscript and aligned it more closely with PLOS One aims and scope.

Thank you for considering our submission. I look forward to the opportunity to contribute to the scholarly dialogue in your esteemed journal. I am available for any further information or clarification that may be required during the review process.

Reviewer #1: General Structure and Rationale

1. Clearly state whether the primary goal is to estimate the prevalence of DRPs or explore associated risk factors. The current phrasing oscillates.

Response to reviewer 1 comment: Thank you for your suggestion. The primary goal is to identify DRPs, assess their prevalence, and examine the factors associated with them. [Addressed in line numbers 81-83, Page 5]

2. The rationale is sound, but why was this particular center (BCH) chosen? More details on selection rationale beyond "governmental and accessible" would help.

Response to reviewer 1 comment: Thank you for your feedback. We have elaborated the rationale. [Addressed in line numbers 95-106, Page 6]

3. The manuscript claims to be the first in Nepal, but doesn't reference enough regional studies to support this. Could there be unpublished or gray literature?

Response to reviewer 1 comment: Thank you for addressing this issue. Most of the studies related to DRPs in breast cancer have been conducted in Kenya and Ethiopia, like countries. There could be a possibility of unpublished or gray literature in our region, which is why we could not cite the articles from our region.

4. There is no reflection on how temporality or causality is limited in the discussion; should be explicitly acknowledged.

Response to reviewer 1 comment: Thank you for highlighting this issue. We have acknowledged our study design limitations in the discussion section. [Addressed in line numbers 337-338, Page 31]

5. Repetition of phrases like “increased DRPs” and “pharmaceutical care” reduces clarity. Consider more varied academic phrasing.

Response to reviewer 1 comment: Thank you for highlighting this context. We have now replaced those words. [Addressed in Line numbers 81-87, Page numbers 5]

Abstract and Keywords

6. The abstract includes excessive numerical detail and minor results; consider summarizing key findings more concisely.

Response to reviewer 1 comment: We have summarized the findings as much as possible. [Addressed in Line numbers 35-43, Page numbers 2-3]

7. Add more specific terms such as “PCNE classification,” “Nepal,” or “oncology pharmacy” for better indexing.

Response to reviewer 1 comment: We have included your suggestions. [Addressed in Line numbers 44-45, Page number 3]

Methods

8. Purposive sampling is a major limitation—state how selection bias was minimized and how representativeness was ensured.

Response to reviewer 1 comment: Thank you for pointing out the limitation. We have introduced the approach so as to minimize the bias in the patient interview, data collection tools, and procedures section. [Addressed in Line numbers 140-145, Page number 8]

9. The initial assumption of 50% prevalence is fine for conservative estimates, but justify why 3 months of data was chosen.

Response to reviewer 1 comment: Thank you for underlining this context. According to hospital records, a total of 497 breast cancer patients (including 10 male patients) visited the hospital in 2079 BS, with the number continuing to rise annually. From Shrawan to Asoj 2080 BS, 121 patients were recorded, indicating an increasing trend. To maximize the sample size, we included data from this period, using a previously published study as a reference (Reference no. 8). [Addressed in Line numbers 125-133, Page number 7]

10. Was the data collection tool (based on PCNE) piloted or validated in this setting?

Response to reviewer 1 comment: Thank you for the valid query. PCNE is a validated tool whose current version, i.e., V9.1, has been developed after a validation round and an expert workshop in February 2020 and is widely accepted. The tool was piloted in the setting. [Addressed in Line numbers 159-163, Page numbers 8-9]

11. Why exclude patients who returned for further chemo? This could skew the sample toward early-stage patients.

Response to reviewer 1 comment: Due to the time constraints, patients were recruited at their first contact regardless of their cancer staging. [Addressed in Line numbers 115-117, Page numbers 6-7]

12. More information is needed on how disagreements were resolved between oncologists. What was the inter-rater reliability?

Response to reviewer 1 comment: Unfortunately, we could not calculate the inter-rater reliability. However, there were a total of 7 disagreements between the two medical oncologists on DRP identification, which were solved through the tumor board meeting. [Addressed in Line numbers 168-176, Page number 9]

13. Were adjustments made for potential confounders (age, comorbidities, etc.)? No multivariable analysis was conducted.

Response to reviewer 1 comment: No adjustment was made for potential confounders, and also, multivariable analysis could not be performed. We have addressed this in the limitations section. [Addressed in Line numbers 372-373, Page number 32

14. Consider at least a basic sensitivity test for key findings due to small sample size.

Response to reviewer 1 comment: Sensitivity analysis was not performed. We have addressed this in the limitations section. [Addressed in Line numbers 375-376, Page number 32]

15. When exactly was DRP assessed—before, during, or after chemo cycles? Clarity would improve reproducibility.

Response to reviewer 1 comment: DRP was assessed after the chemotherapy cycle. [Addressed in Line numbers 168-169, Page number 9]

16. How were ADRs classified—was any scale (Naranjo, CTCAE) used?

Response to reviewer 1 comment: The Severity of the ADRs was not classified using the above scales. However, it was identified based on various assessments. [Addressed in Line numbers 151-152, Page number 8]

17. IRB approval is cited but mention if this study was registered on any clinical trial registry.

Response to reviewer 1 comment: The study was not registered on a clinical trial registry. [Addressed in Line number 186, Page number 10]

Statistical Analysis

18. Only chi-square and Fisher’s exact tests were used. Consider logistic regression to control for multiple variables simultaneously.

Response to reviewer 1 comment: Based on the type of data, chi-square and Fisher’s exact tests were only applicable. Logistic regression could not be performed.

19. Absence of CIs undermines the robustness of prevalence and association measures.

Response to reviewer 1 comment: We agree for this valuable comment. While we agree on the introduction of CIs, our study didn’t include it. We have mentioned this in our limitations section. [Addressed in Line numbers 376-377, Page number 32]

20. Effect sizes (odds ratios, relative risks) are not reported, which makes interpretation of significance less informative.

Response to reviewer 1 comment: We agree on this important part pointed out by the reviewer. Hence, we have addressed this significant part in our limitations section. [Addressed in Line numbers 370-373, Page number 32]

21. Given many comparisons, were p-values adjusted to account for Type I error?

Response to reviewer 1 comment: We appreciate the reviewer’s concern. p-values were not adjusted for multiple comparisons.

Results

22. The 98.9% DRP prevalence is alarmingly high. Is this comparable to similar populations, or could it reflect overclassification?

Response to reviewer 1 comment: Thank you for highlighting the high prevalence of DRPs observed in our study. We agree that a 98.9% prevalence may appear alarmingly high at first glance, and we appreciate the opportunity to clarify its context and methodological basis. To address this concern, we have reviewed comparable studies conducted in similar healthcare settings and populations. Several reports from low-resource or high-risk clinical environments, particularly those involving polypharmacy, limited clinical pharmacy services, or high patient acuity, have documented DRP prevalence rates exceeding 90%. This suggests that our findings, while high, are not unprecedented.

Regarding the possibility of overclassification, we used the [PCNE V9.1], which is a validated and widely accepted framework for identifying and categorizing DRPs. All DRPs were assessed by trained professionals using predefined criteria to minimize subjective interpretation. We acknowledge, however, that classification systems vary in sensitivity, and some may capture a broader spectrum of potential issues. To address this, we have now included a brief discussion in on the potential influence of classification granularity and assessor training on DRP prevalence. We believe the high prevalence reflects the complexity of medication use in our study population rather than overclassification. [Addressed in Line numbers 303-314, Page numbers 29-30]

23. Were any participants excluded due to incomplete data? This is not discussed.

Response to reviewer 1 comment: None of the participants were excluded, as the researcher ensured all of the data set to be collected. [Addressed in Line numbers 154-158, Page number 8]

24. At one point, 104 DRPs in 91 patients are discussed—make clearer what proportion had multiple DRPs.

Response to reviewer 1 comment: Among 92 patients, 84.78 % had one DRP, 14.13% had two DRPs, and 1.1% had none. [Addressed in Line numbers 224-225, Page number 13]

25. Table 1 includes too many levels—merge low-frequency categories (e.g., occupation). Response to reviewer 1 comment: Occupation, number of comorbidities present, and type of chemotherapy regimen had low-frequency categories, which had been updated with corrections in Table 1. [Addressed in Line numbers 216-217, Page numbers 11-13]

26. Reference is made to a figure not shown in the review—ensure figures are labeled and included.

Response to reviewer 1 comment: We have corrected this issue.

Discussion

27. Discussion draws mainly on Ethiopia and Kenya—are there more LMIC examples to support cross-cultural relevance?

Response to reviewer 1 comment: This is a very valid and important point from the reviewer. While we agree on the importance of cross-cultural relevance, DRPs among breast cancer settings are not enough to be discussed from other LMICs because of poor availability and placement of pharmacists in clinical settings. Hence, our major focus was on these countries. We have also added this in our limitations section. [Addressed in Line numbers 380-382, Page numbers 32-33]

28. Phrases such as “DRPs were caused by…” should be tempered due to study design limits.

Response to reviewer 1 comment: We have corrected the sentence formatting throughout the paper.

29. Expand on why timing/dosing issues are so prevalent—does this point to health literacy gaps, regimen complexity, or system failures?

Response to reviewer 1 comment: We have included the reason for timing/dosing issues. [Addressed in Line numbers 330-332, Page number 30]

30. High number of drugs prescribed is noted—discuss whether this is guideline-recommended or potentially excessive.

Response to reviewer 1 comment: We have included the clarification for overprescribing. [Addressed in Line numbers 297-302, Page number 29]

31. Many intervention ideas are proposed, but no feasibility or pilot data are cited—frame as hypotheses or recommendations.

Response to reviewer 1 comment: We have now added several recommendations within the discussion and strength and limitation sections where applicable.

32. Good point raised—but explain if Nepal’s national PV system supports hospital-based units practically.

Response to reviewer 1 comment: Though the Department of Drug Administration of Nepal supports the hospital-based PV units, due to resource constraints, its practical implication is weak. [Addressed in Line numbers 324-325, Page number 30]

33. Does BCH have a formal reporting mechanism for ADRs? Mention if findings have been/will be reported there.

Response to reviewer 1 comment: No, BCH does not have a formal reporting system for ADR. However, ADRs are managed by the consultants. At the time of study, BCH was not a regional PV centre, but it has now been designated as one of them. [Addressed in Line numbers 325-328, Page number 30]

Language and Style

34. Past and present tenses are used inconsistently throughout the methods and results.

Response to reviewer 1 comment: Thank you for the suggestions. The sentence structure has been changed throughout the paper.

35. Phrases like “it might be difficult to generalize the findings” occur multiple times. Use once and move on.

Response to reviewer 1 comment: Thank you for reminding. We have corrected this issue.

36. Numerous minor grammatical errors (e.g., "do not explain" should be "does not explain")—recommend copyediting pass.

Response to reviewer 1 comment: We have edited the paper through Grammarly.

37. "Protein powder," "immunity booster"—provide generic names or ingredient lists to ensure scientific precision.

Response to reviewer 1 comment: We have added the ingredients list for protein powder and immunity booster in Table 3. [Addressed in Line numbers 234-235, Page numbers 15-24]

38. Terms like “TAC” and “FEC” appear before being defined—define all chemotherapy abbreviations upon first use.

Response to reviewer 1 comment: We have made corrections to this issue. [Addressed in Line numbers 219 and 234, Page numbers 15 and 16 respectively]

Conclusion

39. Statements about fulfilling National Cancer Control Strategy goals are too broad. Keep the focus on DRP assessment and interventions.

Response to reviewer 1 comment: We have deleted this broad goal and made it more specific. [Addressed in Line numbers 392-397, Page number 33]

40. Mention specific designs (e.g., prospective cohort, pharmacist-led intervention RCTs) instead of vague calls for “more studies.

Response to reviewer 1 comment: We have included your suggestion. [Addressed in Line numbers 396-397, Page number 33]

Reviewer #2: So far my comments has been addressed but I highly recommend the authors to use some sort of tools to improve the understanding of the sentences in manuscript. I still stick with the comment for revising the writing/structure of the sentences throughout the manuscript to make it unambiguous with appropriate flow.

Response to reviewer 2 comment: Thank you for the constructive feedback. We have addressed your comment on revising the writing/structure of the sentences, which has been performed throughout the manuscript.

---

## [Decision Letter · Decision Letter 2]

1 Oct 2025

Assessment of drug-related problems among breast cancer patients in a cancer specialty center in Nepal

PONE-D-24-58527R2

Dear Dr. Sah,

We’re pleased to inform you that your manuscript has been judged scientifically suitable for publication and will be formally accepted for publication once it meets all outstanding technical requirements.

Kind regards,

Vijayaprakash Suppiah, PhD

Academic Editor

PLOS ONE

**Comments to the Author**

1. If the authors have adequately addressed your comments raised in a previous round of review and you feel that this manuscript is now acceptable for publication, you may indicate that here to bypass the “Comments to the Author” section, enter your conflict of interest statement in the “Confidential to Editor” section, and submit your "Accept" recommendation.

Reviewer #2: All comments have been addressed

2. Is the manuscript technically sound, and do the data support the conclusions?

Reviewer #2: Yes

3. Has the statistical analysis been performed appropriately and rigorously? 

Reviewer #2: Yes

4. Have the authors made all data underlying the findings in their manuscript fully available?

Reviewer #2: Yes

5. Is the manuscript presented in an intelligible fashion and written in standard English?

Reviewer #2: Yes

6. Review Comments to the Author

Reviewer #2: The revised manuscript looks good for the publication. However, I notice no confidence interval reported. Only p-values are reported. CIs might give a better sense of the precision of the findings to the readers.

7. PLOS authors have the option to publish the peer review history of their article (what does this mean? ). If published, this will include your full peer review and any attached files.

**Do you want your identity to be public for this peer review?** For information about this choice, including consent withdrawal, please see our Privacy Policy .

Reviewer #2: No

---

## [Editor Report · Acceptance letter]

PONE-D-24-58527R2

PLOS ONE

Dear Dr. Sah,

I'm pleased to inform you that your manuscript has been deemed suitable for publication in PLOS ONE. Congratulations! Your manuscript is now being handed over to our production team.

Kind regards,

on behalf of

Dr. Vijayaprakash Suppiah

Academic Editor

PLOS ONE